# Effects of Nursing Work Environment, Need Satisfaction, and Depression on Turnover Intention in Korea

**DOI:** 10.3390/healthcare11121698

**Published:** 2023-06-09

**Authors:** Sun-Hwa Shin, On-Jeon Baek, Eun-Hye Lee

**Affiliations:** Nursing Department, College of Nursing, Sahmyook University, Seoul 01795, Republic of Korea; shinsh@syu.ac.kr (S.-H.S.); perfectoj@naver.com (O.-J.B.)

**Keywords:** nurse, work environment, personnel turnover, personal satisfaction, depression

## Abstract

This study examined the effects of the nursing work environment, need satisfaction, and depression on turnover intention in South Korean nurses through a mediating model. This study was a descriptive cross-sectional research study and was conducted through an online questionnaire. A total of 248 nurses were recruited for this study. Data were collected in August 2022. Participants were invited to complete self-reported questionnaires that measure nursing work environment, need satisfaction, depression, turnover intention, and demographic information. The data obtained were analyzed using the dual mediation model applying the PROCESS macro (Model 6). This study analyzed the direct effects of the nursing work environment on need satisfaction, depression, and turnover intention. The nursing work environment also had indirect effects on their turnover intention via need satisfaction and depression. The mediating effect of need satisfaction, which affects the turnover intention by increasing the satisfaction of the nursing work environment, was found to be the greatest. It has been shown that the more positive a nurse’s experience of the nursing work environment, the higher the nurse’s need satisfaction. According to the study results, the increase in nurses’ need satisfaction greatly contributes to the decrease in depression and turnover intention. Therefore, active efforts should be made to improve the nursing work environment to fulfill basic needs.

## 1. Introduction

Nurses are required to work three shifts due to the nature of their job, which requires them to care for inpatients for 24 h. Due to the severity of patients’ illnesses and specialized healthcare demands, nurses experience high levels of stress during their shifts [1]. According to the Yerkes-Dodson laws, high levels of arousal due to stress can lead to emotional burnout and decreased performance when conducting highly cognitive nursing tasks [2]. Moreover, emotional burnout has also been reported to be the most influential variable affecting turnover intention [3,4]. Nurses are the most important workforce in healthcare organizations. The negative impact of increased nurse turnover on the quality of patient care has also been reported [5].

Nurses often face situations in which their basic needs are unmet while providing high-intensity nursing care. A survey conducted by the Korean Nurses Association in 2018 examining human rights violations impacting nurses revealed various unsatisfied physiological needs, such as not being guaranteed mealtimes (31.1%) or rest times (54.4%), as well as unmet safety needs, such as protections against verbal abuse (65.5%) and assault (10.0%) [6]. “Needs” refers to the desire to obtain or do something [7]. Human needs are closely related to motivation and behavior and play an essential role in human growth and development [8]. External factors, such as the working environment, tasks, and organizational culture, influence nurses’ needs and may or may not be met.

Nurses’ perceptions of their working environment influence turnover intention [9]. The American Nurses Credentialing Center proposes standards for nursing staff and physical resources to provide quality care and awards “Magnet Nursing Services Recognition” to healthcare institutions that meet them [10]. A key characteristic of accredited institutions is that nurses perceive their work environment positively [10,11,12]. Accreditation programs in the United States have been proposed as an alternative to reduce nurse turnover and ensure sufficient nursing staff [12]. On the other hand, Korea does not have an accreditation system to secure nursing staff. Moreover, creating a favorable nursing work environment has resulted in positive outcomes for both nurses and patients [12,13]. In other words, favorable nursing work environments have resulted in nurses’ increased participation and improved job satisfaction [13,14]. Previous studies of nurses in various fields, such as nursing managers, new nurses, and hospice nurses, have shown that the more positively nurses perceive their nursing work environment in healthcare institutions, the lower their turnover intention [15,16,17].

Depression in nurses is closely related to job stress, which is a consistent ability to deal effectively with stress and psychological problems and to rationally resolve conflict situations [1]. Depression directly affects nurse turnover intention [18] and has been strongly linked to working environments at night and during shifts [19]. Previous studies have shown that the more negatively nurses perceive their working environment, the greater the negative impact on their mental health [20]. As such, it was found that nurses’ work environments affect their mental health [20], resulting in issues such as depression and increased turnover intention [9,10,21].

The more nurses’ needs have been satisfied, the lower the turnover intention, suggesting that it is essential to improve nurses’ satisfaction during work [22]. Based on study findings that satisfying an individual’s basic psychological needs can reduce mental health problems such as depression [23], it can be inferred that fulfilling nurses’ needs will alleviate depression. However, there is a lack of evidence to confirm the relationships between nursing work environment and nursing need satisfaction and between depression and need satisfaction.

Therefore, this study aimed to identify the relationships between nursing work environment, need satisfaction, depression, and turnover intention among nurses. The results of this study will provide essential data for creating a healthy nursing working environment that promotes the physical, mental, and social health of nurses and for preparing measures to reduce nurses’ turnover intention.

The purpose of this study is to examine the relationship between the nursing work environment, need satisfaction, depression, and turnover intention for nurses, for which the following findings were revealed: First, the correlation between nursing work environment, need satisfaction, depression, and turnover intention was confirmed. Second, the mediating effect of need satisfaction was confirmed in the relationship between nursing work environment and turnover intention. Third, the mediating effect of depression was confirmed in the relationship between nursing work environment and turnover intention. Fourth, the double mediating effect of need satisfaction and depression was confirmed in the relationship between nursing work environment and turnover intention.

## 2. Materials and Methods

### 2.1. Study Design

This descriptive cross-sectional study examined the correlation between the nursing work environment, need satisfaction, depression, and turnover intention among South Korean nurses.

### 2.2. Participants

The study participants were practicing nurses in South Korea working in 300+ bed general hospitals in Seoul and Gyeonggi Province. The inclusion criteria were nurses working three shifts, performing direct nursing care for patients, and exhibiting no cognitive impairments that would prevent them from reading and responding to questionnaires. Full-time nurses, those not performing direct nursing care for patients as their main job, and nurses who had limitations in reading and responding to the questionnaire were excluded. The number of cases required for this study was calculated using the G*power 3.1 program, and the regression analysis revealed a moderate effect size (0.15), Type I error of 0.05, power of 0.95, and a total of ten independent variables. The minimum number of cases was calculated as 172. Considering the dropout rate, a survey was conducted with 250 nurses, and data were finally collected from 248 nurses to meet the minimum number of cases.

### 2.3. Research Instruments

#### 2.3.1. Nursing Work Environment

Nursing work environment was measured using the Korean version of the Practice Environment Scale of the Nursing Work Index (K-PES-NWI) developed by Lake [12], whose reliability and validity were verified by Cho et al. [24]. The nursing work environment measurement tool consists of twenty-nine questions and is divided into five sub-domains (participation of nurses in hospital operation, foundation for quality nursing, nursing manager’s ability, leadership, support for nurses, sufficient manpower and physical support, and cooperation between nurses and doctors). Responses to the questions were measured on a 4-point Likert scale ranging from 1 = “strongly disagree” to 4 = “strongly agree”. The scores ranged from 29 to 116 points, with higher total scores indicating more positive experiences of the nursing work environment. Cronbach’s α was 0.93 in the study by Cho et al. [24] and 0.92 in this study.

#### 2.3.2. Need Satisfaction

Need satisfaction was measured using the nurse need satisfaction measurement tool developed by Kim and Shin [25], which was based on Maslow’s hierarchy of needs theory. The need satisfaction measurement tool consists of thirty questions, each comprising six questions for each of the five needs (physiological, safety, belonging, respect, and self-realization needs). Responses to the questions were measured on a 5-point Likert scale ranging from 1 = “strongly disagree” to 5 = “strongly agree”, and the physiological needs items were reverse-coded. Scores ranged from 30 to 150, with higher total scores denoting a higher satisfaction with needs. Cronbach’s α was 0.90 in the study by Kim and Shin [25] and 0.92 in this study.

#### 2.3.3. Depression

Depression was measured using the Brief Symptoms Inventory-18 (BSI-18) developed by Park et al. [26] for college students. It consists of depression (six questions), anxiety (four questions), physicalization (five questions), and panic (three questions). Only the six questions related to depression were used in this study. Responses to the questions were measured on a 5-point Likert scale ranging from 1 = “almost never” to 5 = “almost always”. Scores ranged from 6 to 30, with higher total scores indicating higher levels of depression. Cronbach’s α value for depression was 0.89 in the study by Park et al. [26] and 0.83 in this study.

#### 2.3.4. Turnover Intention

Turnover intention was measured using the turnover intention measurement tool developed by Mobley [27] and modified by Moon [28]. Turnover intention was measured via five questions and responses were measured on a 5-point Likert scale ranging from 1 = “not at all” to 5 = “very much”. Scores ranged from 5 to 25, with higher total scores denoting higher degrees of turnover intention. Cronbach’s α was 0.85 in Moon’s [28] study and 0.79 in this study.

### 2.4. Data Collection

This study deliberated on the research procedures and ethical considerations at the Institutional Review Board of Sahmyook University to which the researchers belong (IRB No: SYU 2022-04-011). After submitting to a research ethics review, data were collected from 16 August to 30 August 2022 via simultaneous offline and online surveys. For the offline survey, two research assistants were trained in advance on the purpose and procedure of the study, and data were collected. The research director visited the nursing department of a general hospital in Seoul, explained the purpose of the study, obtained permission to conduct it, and posted a recruitment notice for the study participants. The research assistant explained the purpose of the study to nurses who voluntarily expressed their intention to participate, ensured that all participants signed consent forms, and distributed the questionnaire. After collecting the completed questionnaire, the consent forms and questionnaires were stored in separate envelopes. The online survey was conducted with nurses working at general hospitals with more than 300 beds in Seoul and Gyeonggi-do. E-mails with the link address were sent to request their voluntary participation in the online survey. The online questionnaire was created using Google Forms and was designed so that the questionnaire could only be started after the participants had read the explanation and voluntarily agreed to participate. The survey was conducted anonymously, and no personal or sensitive information was collected. Furthermore, participants were informed that there were no disadvantages due to abstaining from the survey or quitting halfway. Gift certificates were provided to those who completed the offline or online surveys.

### 2.5. Data Analyses

Data analysis was performed using SPSS (version 26.0; IBM Institute, Albany, NY, USA). Descriptive statistics of frequency, percentage, mean, and standard deviation were calculated for the participants’ general characteristics. Differences in nursing working environment, need satisfaction, mental health, and turnover intention according to general characteristics were analyzed using an independent *t*-test and one-way analysis of variance (ANOVA), and post hoc tests were performed using a Scheffé method. Pearson’s correlation coefficient analysis was used to analyze the relationship between the main study variables. The mediating effect was analyzed using PROCESS Macro (Model 6), and the 95% confidence interval (CI) was calculated using the bootstrapping method to verify the significance of the mediating effect size and the difference by the mediating effect path.

## 3. Results

### 3.1. Participant Characteristics

Regarding respondents’ general characteristics (Table 1), 219 participants were women (88.3%) and the average age was 27.54 years (±3.49) with 193 women in their 20s (77.8%). As for marriage status, 219 (88.3%) were unmarried. In terms of education level, 229 (92.3%) had graduated from a four-year university. Regarding religion, 155 nurses (62.5%) responded that they had no religion. As for working experience, 102 (41.1%) had 25 to 60 months of experience. A total of 175 participants (70.6%) were in general hospitals, and the number of beds in hospitals was 154 participants (62.1%) with less than 300 to 499 beds. A total of 121 participants (48.8%) reported working in the internal medicine ward, and 116 (46.8%) answered that they perceived they were healthy.

### 3.2. Differences in Nursing Working Environment, Need Satisfaction, Depression, and Turnover Intention According to General Characteristics

Table 1 shows the results of analyzing the differences in the nursing working environment, need satisfaction, depression, and turnover intention according to participants’ general characteristics. There was a significant difference in the nursing work environment according to career (t = 7.48, *p* = 0.001) and perceived health status (t = 4.77, *p* = 0.009). As a result of the career post-test, participants who worked for 6 to 24 months had significantly higher scores on the nursing environment than those who worked for 25 to 60 months and for 61 months or more. Need satisfaction differed significantly according to education (t = 5.29, *p* < 0.001), religion (t = −1.99, *p* = 0.047), and perceived health status (t = 23.90, *p* < 0.001). The results of the post hoc test revealed that participants who graduated from a four-year university had significantly higher scores on need satisfaction than those who graduated from graduate school. Regarding perceived health status, more participants responded that they were “healthy” than “moderate”, and participants who responded “moderate” had significantly higher need satisfaction scores than participants who responded “unhealthy”. Depression differed significantly according to the ward (t = 2.84, *p* = 0.038) and perceived health status (t = 9.81, *p* < 0.001). The post hoc test revealed no significant difference in the depression scores by ward. Regarding their perceived health status, participants who answered “unhealthy” had significantly higher depression scores than those who reported being “moderate” or “healthy”. There was a significant difference in turnover intention according to perceived health status (t = −5.77, *p* = 0.004). The post hoc test showed that nurses who indicated they were “unhealthy” had significantly higher turnover intention scores than those who responded “moderate” or “healthy”.

### 3.3. Correlations among the Nursing Work Environment, Need Satisfaction, Mental Health, and Turnover Intention

The average scores of the major study variables (Table 2) were 2.61 points (±0.40) for nursing working environment, 2.95 points (±0.51) for need satisfaction, 2.42 points (±0.83) for mental health, and 3.60 points for turnover intention (±0.79). An analysis of the correlation between the nursing working environment, need satisfaction, depression, and turnover intention (Table 2) revealed that the nursing work environment had a positive correlation with need satisfaction (r = 0.72, *p* < 0.001), a positive correlation with depression (r = −0.33, *p* < 0.001), and a significant negative correlation with turnover intention. Need satisfaction had a significant negative correlation with depression (r = −0.50, *p* < 0.001) and turnover intention (r = −0.49, *p* < 0.001). Finally, depression was positively correlated with turnover intention (r = 0.34, *p* < 0.001).

### 3.4. Significance of the Mediating Effect between Need Satisfaction and Depression and Difference in the Mediating Effect

Table 3 shows the results of confirming the double mediating effect of need satisfaction and depression in the relationship between the nursing work environment and turnover intention. Among the general characteristics, the mediating model analysis was performed after including perceived health status, which showed a significant difference in turnover intention as a control variable. In Model 1, the independent variable of the nursing working environment had a significant positive effect on the primary parameter (M1) and need satisfaction (β = 0.66, *p* < 0.001), and the explanatory volume of the model was 58.6% (R^2^ = 0.586, F = 173.38, *p* < 0.001). In Model 2, the nursing working environment did not have a significant effect on the secondary parameter (M2) of depression (β = 0.04, *p* = 0.658), and need satisfaction (M1) had a significant negative effect on the secondary parameter of depression (β = −0.49, *p* < 0.001). The explanatory power of Model 2 was 25.8% (R^2^ = 0.258, F = 28.27, *p* < 0.001). In Model 3, in which the two parameters were simultaneously input, need satisfaction (M1) had a significant negative effect on turnover intention (Y) (β = −0.44, *p* < 0.001), and depression (M2) had a significant positive effect on turnover intention (Y) (β = 0.13, *p* = 0.044). Additionally, it was found that the nursing work environment did not affect turnover intention when the parameters of satisfaction of desire and depression were input simultaneously (β = 0.01, *p* = 0.907), The explanatory power of the model was 25.1% (R^2^ = 0.251, F = 20.32, *p* < 0.001).

The significance of the mediating effect of need satisfaction and depression on the relationship between the nursing working environment and turnover intention was verified with a 95% CI using the bootstrapping method (Table 4). The mediating effect of the nursing working environment on turnover intention through need satisfaction (Indirect 1) was statistically significant (B = −0.58, Boot 95% CI [−0.85, −0.30]). However, the mediating effect of the nursing working environment on turnover intention through depression (Indirect 2) was not statistically significant (B = 0.01, Boot 95% CI [−0.04, 0.06]). Finally, the mediating effect of the nursing working environment on turnover intention through the double mediation of satisfaction and depression (Indirect 3) was statistically significant (B = −0.08, Boot 95% CI [−0.19, −0.01]).

The difference in the mediating effect of each of the three pathways was verified. In the process of the nursing work environment affecting turnover intention, the mediating effect of need satisfaction (Indirect 1) was significantly different from the mediating effect of depression (Indirect 2) (ΔB = −0.59, 95% CI [−0.86, −0.32]). In addition, the mediating effect of satisfaction of desire (Indirect 1) was significantly different from the dual mediating effect of satisfaction of desire and depression (Indirect 3) (ΔB = −0.05, 95% CI [−0.81, −0.17]). In other words, the mediating effect of the nursing work environment affecting turnover intention by increasing need satisfaction (B = −0.58) stems from the fact that the nursing work environment affects turnover intention by reducing depression (B = 0.01), and the nursing work environment was found to be significantly higher than the mediating effect (B = −0.08) that impacts turnover intention by changing need satisfaction and depression. Through this, it was found that the mediating effect of need satisfaction was strong within the relationship between the nursing work environment and turnover intention (Figure 1).

## 4. Discussion

This study analyzed the mediating effects to examine the relationships among the nursing work environment, need satisfaction, depression, and turnover intention among currently employed nurses. Based on these findings, we discuss the following points.

First, we found that the nursing work environment experienced by nurses had a direct impact on turnover intention. Previous studies have also found that nursing work environment significantly affects nurses’ turnover intention [9,21,29]. Thus, our study results are consistent with those of previous studies confirming the direct effect of the nursing work environment on turnover intention. In an earlier study, experiences of nursing work environment were, on average, 2.97 out of 5 [30] and 2.81 out of 5 [20]. In this study, it was 2.61 out of 5, thus confirming that the experience of the nursing work environment was more negative than those in previous research. Improving nursing work environment has contributed to increasing patient safety awareness and improving nurses’ quality of nursing care and job satisfaction [31]. Therefore, efforts to improve the nursing work environment would have significantly enhanced the quality of care.

Nonetheless, this study highlights that when the mediating variables of need satisfaction and depression were added, the direct effect of the nursing work environment on turnover intention was not significant. In other words, it is essential that this study has found a completely mediated impact of need satisfaction and depression in influencing turnover intention in the nursing work environment. A previous study discovered that limited opportunities for promotion, conflicts among colleagues, low salaries, and poor security increased nurses’ turnover intention [32]. It can be considered that no matter how beneficial the nursing work environment, if the need satisfaction that nurses desire is poorly addressed, it will be difficult to lower their turnover intention.

The mediating effect of need satisfaction on the relationship between nurses’ work environment and turnover intention was significant. There are not enough studies investigating the relationship between nursing work environment and need satisfaction to make a direct comparison. However, nurses’ need satisfaction was found to have a significant effect on turnover intention [22], which partially supports the results of this study. The physical nursing environment, such as shift work and night work, had a significant effect on the physical and emotional state of nurses [33]. Unsafe nursing work environments have been shown to cause nurse burnout and increase turnover intention [34], which is consistent with the findings of this study. Maslow’s hierarchy of needs theory suggests five basic human needs: physiological needs, safety needs, belongingness needs, esteem needs, and self-actualization needs [8]. In order to create a healthy nursing work environment, it is necessary to carefully examine and study the five basic needs so that they can be satisfied in a balanced manner.

The dual mediating effects of need satisfaction and depression were significant in the relationship between nurses’ work environment and turnover intention. This study confirmed that the indirect effects of need satisfaction and depression are greater than the direct effects of nursing work environment on turnover intention, which is meaningful. In other words, the degree to which nurses experience positive or negative nursing work environment affects need satisfaction, and need satisfaction affects depression, which changes turnover intention. This is partially consistent with the findings of a previous study [35] which found that while turnover intention may be lower in favorable nursing work environments, there is a relationship between nursing work environment and turnover intention only when nurses’ job satisfaction is high. Increasing nurses’ job satisfaction through improved working conditions promotes nurse retention [35]. Job satisfaction was found to increase when nurses’ psychological or physical needs with guaranteed rest time were met [36]. Satisfaction with social needs through the support of colleagues and superiors was also positively correlated with job satisfaction [36]. In addition, nurses’ psychological satisfaction reduced fatigue and turnover intention, improved patient care, and increased productivity [37]. To reduce nurses’ turnover intention, it is necessary to reduce depression and improve the quality of nursing work [38]. Therefore, it is necessary to establish a nursing workforce management system that periodically monitors nurses’ satisfaction and depression and applies intervention programs to manage need satisfaction and depression.

The mediating effect of depression was insignificant in the relationship between the nursing work environment and turnover intention. This result means that nurses’ negative experiences of the nursing work environment did not increase depression. The results showed a negative correlation between the nursing work environment and depression. Nevertheless, the effect of the nursing work environment on depression decreased as the need satisfaction variable was added. This finding contradicts those of previous studies [9,20,21] which reported that the nursing work environment significantly affects mental health problems. Therefore, repeated studies must be conducted to re-examine whether this is due to differences in the measurement tools for depression.

The limitations and suggestions of this study are as follows: First, it included nurses working at general hospitals in Seoul. However, there are limitations in that it did not consider the size of the hospital, regional characteristics, organizational culture, and the workforce management system. Therefore, to generalize and ensure the validity of this study’s results, repeated studies should be conducted considering the characteristics of various regions and hospital levels. Second, there is a limitation to inferring causal relationships. This study used a cross-sectional design to examine the relationship between nursing work environment, need satisfaction, depression, and turnover intention. Therefore, we propose proving causality through studies that identify the reduction in turnover intention by developing and applying programs that promote the nursing work environment and need satisfaction. Finally, we suggest a follow-up study to examine the relationship between the work environment, need satisfaction, depression, and turnover intention for nurses and various occupational groups working in medical institutions and to compare the relationships between groups.

## 5. Conclusions

This study examined the effects of the nursing work environment, need satisfaction, and depression on turnover intention in nurses through a mediating model. As a result of this study, it was found that the mediating effect of need satisfaction, which affects turnover intention in the nursing work environment, was the strongest factor. In addition, the mediating effect of depression in the nursing work environment reduced turnover intention, and the dual mediating impacts of need satisfaction and depression on the relationship between the nursing work environment and turnover intention were both significant. Considering that the more positive the experience of the nursing work environment, the higher the influence of increasing need satisfaction, it is essential to create a healthy nursing work environment. In addition, since the increased need satisfaction of nurses contributes significantly to reducing depression and turnover, efforts should be made to improve nursing work to find ways to address basic needs.

## Figures and Tables

**Figure 1 healthcare-11-01698-f001:**
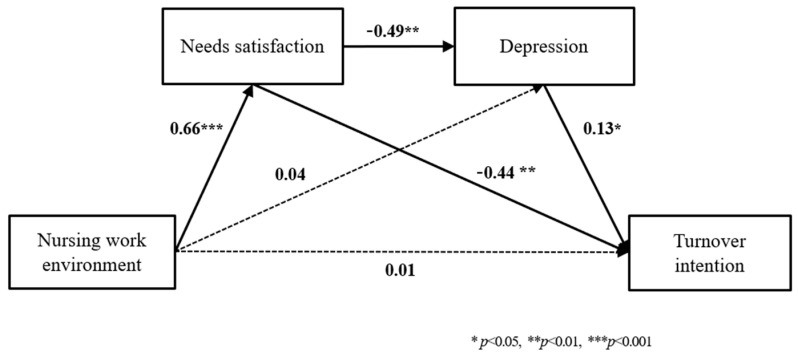
Research model.

**Table 1 healthcare-11-01698-t001:** Differences in Nursing Working Environment, Need Satisfaction, Depression, and Turnover Intention According to General Characteristics (N = 248).

Variables	Categories	Nursing Work Environment		Need Satisfaction		Depression		Turnover Intention	
		M ± SD	t/F (*p*)	M ± SD	t/F (*p*)	M ± SD	t/F (*p*)	M ± SD	t/F (*p*)
Gender	Male	2.57 ± 0.33	−0.49 (0.626)	3.04 ± 0.42	1.04 (0.298)	2.22 ± 0.71	−1.38 (0.169)	3.48 ± 0.79	−0.86 (0.393)
	Female	2.61 ± 0.41		2.94 ± 0.52		2.44 ± 0.84		3.62 ± 0.79	
Age	20 yrs	2.62 ± 0.40	0.82 (0.415)	2.94 ± 0.48	−0.42 (0.677)	2.41 ± 0.81	−0.44 (0.661)	3.58 ± 0.79	−0.87 (0.385)
	≥30 yrs	2.57 ± 0.38		2.98 ± 0.60		2.46 ± 0.89		3.68 ± 0.82	
Marital status	Unmarried	2.61 ± 0.40	0.74 (0.458)	2.94 ± 0.49	−0.66 (0.507)	2.43 ± 0.82	0.46 (0.645)	3.58 ± 0.78	−1.29 (0.200)
	Married	2.55 ± 0.37		3.01 ± 0.62		2.35 ± 0.92		3.78 ± 0.88	
Educational background	Three-year university ^a^	2.61 ± 0.49	0.33 (0.716)	2.94 ± 0.56	5.28 (0.006) b < c	2.27 ± 0.79	0.76 (0.468)	3.70 ± 0.66	0.11 (0.896)
	Four-year university ^b^	2.60 ± 0.40		2.93 ± 0.49		2.44 ± 0.83		3.60 ± 0.80	
	Over graduate school ^c^	2.71 ± 0.31		3.48 ± 0.51		2.13 ± 0.75		3.53 ± 0.81	
Religion	No religion	2.60 ± 0.39	−0.26 (0.795)	2.90 ± 0.50	−1.99 (0.047)	2.47 ± 0.86	1.21 (0.226)	3.65 ± 0.80	1.15 (0.253)
	Religious	2.61 ± 0.41		3.03 ± 0.50		2.34 ± 0.77		3.53 ± 0.77	
Career	6~24 months (≤2 yrs) ^a^	2.75 ± 0.42	7.48 (0.001) a > b,c	3.00 ± 0.44	1.05 (0.350)	2.26 ± 0.78	1.91 (0.150)	3.48 ± 0.75	1.29 (0.277)
	25~60 months (3~5 yrs) ^b^	2.52 ± 0.34		2.90 ± 0.47		2.50 ± 0.82		3.66 ± 0.79	
	≥61 months (≥5 yrs) ^c^	2.58 ± 0.41		2.97 ± 0.60		2.46 ± 0.86		3.65 ± 0.84	
Hospital type	University hospital	2.55 ± 0.44	−1.46 (0.147)	3.03 ± 0.56	1.45 (0.151)	2.32 ± 0.89	−1.26 (0.210)	3.56 ± 0.95	−0.52 (0.606)
	General hospital	2.63 ± 0.38		2.92 ± 0.48		2.46 ± 0.80		3.62 ± 0.72	
Number of beds in hospital	300~499 beds	2.63 ± 0.38	1.02 (0.307)	2.92 ± 0.46	−1.15 (0.250)	2.47 ± 0.79	1.22 (0.222)	3.62 ± 0.71	0.33 (0.742)
	≥500 beds	2.57 ± 0.43		3.00 ± 0.57		2.34 ± 0.88		3.58 ± 0.92	
Working department	Internal medicine ward	2.55 ± 0.41	1.74 (0.160)	2.89 ± 0.53	2.52 (0.059)	2.55 ± 0.86	2.84 (0.038)	3.70 ± 0.81	1.56 (0.200)
	Surgery medicine ward	2.65 ± 0.39		2.96 ± 0.47		2.39 ± 0.80		3.53 ± 0.77	
	Special department	2.63 ± 0.35		3.02 ± 0.41		2.21 ± 0.70		3.53 ± 0.75	
	Miscellaneous	2.76 ± 0.42		3.23 ± 0.58		2.08 ± 0.85		3.34 ± 0.85	
Perceived health status	Healthy ^a^	2.67 ± 0.35	4.77 (0.009) a > c	3.13 ± 0.44	23.90 (<0.001) a > b > c	2.20 ± 0.81	9.81 (<0.001) c > a,b	3.50 ± 0.81	5.77 (0.004) c > a,b
	Moderate ^b^	2.58 ± 0.42		2.86 ± 0.45		2.55 ± 0.77		3.61 ± 0.79	
	Unhealthy ^c^	2.41 ± 0.47		2.46 ± 0.64		2.90 ± 0.90		4.10 ± 0.53	

a,b,c Alphabets refer to post hoc test results using Scheffé’s method.

**Table 2 healthcare-11-01698-t002:** Correlations among the Nursing Work Environment, Need Satisfaction, Depression, and Turnover Intention (N = 248).

Variables	Need Satisfaction	Depression	Turnover Intention	Mean ± SD	Skewness	Kurtosis
	r (*p*)					
Nursing work environment	0.72 (<0.001)	−0.33 (<0.001)	−0.34 (<0.001)	2.61 ± 0.40	−0.01	0.45
Need satisfaction		−0.50 (<0.001)	−0.49 (<0.001)	2.95 ± 0.51	−0.28	1.40
Depression			0.34 (<0.001)	2.42 ± 0.83	0.30	−0.42
Turnover intention				3.60 ± 0.79	−0.38	−0.13

**Table 3 healthcare-11-01698-t003:** Results of Mediating Effect Analysis (N = 248).

Model	DV	IV	B	SE	β	t	*p*	Adj. R^2^	F (*p*)
1	M1	X	0.84	0.05	0.66	15.87	<0.001	0.586	173.38 (<0.001)
2	M2	X	0.07	0.17	0.04	0.44	0.658	0.258	28.27 (<0.001)
		M1	−0.81	0.14	−0.49	−5.76	<0.001		
3	Y	X	0.02	0.16	0.01	0.12	0.907	0.251	20.32 (<0.001)
		M1	−0.69	0.14	−0.44	−4.77	<0.001		
		M2	0.13	0.06	0.13	2.03	0.044		

X = Nursing work environment; M1 = Need satisfaction; M2 = Depression; Y = Turnover intention; DV = Dependent variable; IV = Independent variable; Adjusted for perceived health status.

**Table 4 healthcare-11-01698-t004:** Significance Test of Mediating Effects of Need Satisfaction and Depression (N = 248).

Model	Variables	Direct Effect				Indirect Effect			
		Effect	SE	95% CI		Effect	SE	95% CI	
				Boot LLCI	Boot ULCI			Boot LLCI	Boot ULCI
Direct	X→Y	−0.66	0.14	−0.93	−0.39				
Indirect 1	X→M1→Y					−0.58	0.14	−0.85	−0.30
Indirect 2	X→M2→Y					0.01	0.02	−0.04	0.06
Indirect 3	X→M1→M2→Y					−0.08	0.05	−0.19	−0.01
Differences (ΔB)	Indirect 1-Indirect 2					−0.59	0.14	−0.86	−0.32
	Indirect 1-Indirect 3					−0.50	0.16	−0.81	−0.17
	Indirect 2-Indirect 3					0.09	0.06	0.01	0.24

X = Nursing work environment; M1 = Need satisfaction; M2 = Depression; Y = Turnover intention; CI = confidence interval; LLCI = lower-level confidence interval; ULCI = upper-level confidence interval.

## Data Availability

The datasets used and/or analyzed during the current study are available from the corresponding author on reasonable request.

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
