# Peer review of "Effects of Nursing Work Environment, Need Satisfaction, and Depression on Turnover Intention in Korea"

_healthcare, 2023, doi:10.3390/healthcare11121698_

Round 1
Reviewer 1 Report
Dear Authors.
This paper needed a lot of work. Several suggestions to follow.
Repetitions: f.e. „were” x 3 in line 11 “satisfaction”, “needs” (line 21,22) etc. Be so kind and reconstruct phrases. Please check Your paper in that matter.
This phrase is syntax unclear also….
Lines 29 – 30 “management” of what (institutions or workforce)?
Iine 28 – stress is encouraging too. See, f.e. Yerkes-Dodson laws ….[http://dlibra.pbs.edu.pl/Content/448/PDF/janowski_2013_personality_matter.pdf]
Edward J. Calabrese, Converging concepts: Adaptive response, preconditioning, and the Yerkes–Dodson Law are manifestations of hormesis RSS, Ageing Research Reviews 2008, 7(1), 1, 8-20
I strongly recommend to mention above papers to clarify Your point of view.
Lines 31-34 – It would be very significant to recognize what the work description, during recruitment, was……
Line 49 source [12] what is the reason and common dominator of Korean and US nurses?
Line 105. The subject of conducted research (research sample) were nurses or patients?
2.3.1. The Likert scale should be odd – You cannot reach a median otherwise
Line 164. What about computer IP?
Kind Regards
This paper should be checked by native speaker. Too many repetitions.
Author Response
Dear Editor and Reviewers,
We wish to thank you for your thoughtful comments and valuable feedback on the manuscript originally titled, “Effects of Nursing Work Environment, Need Satisfaction, and Depression on Turnover Intention in Korea”.
We have modified the manuscript according to your suggestions, rewriting and rephrasing sections to improve clarity, adding further information, and explaining in detail the points that were previously vague. For your convenience, we have set the revisions in the manuscript in red. We believe that the revised version of this paper will be of interest to the readership of the Healthcare.
We have seriously considered the reviewers’ comments and carefully revised the manuscript. Please find our response on the attached document.

Reviewer 2 Report
I think it is a valuable study because you analyzed and presented the data elaborately so that we can find a strategy to lower turnover intention of nurses. However, I would like to give you a review opinion below, so please consider it. Thank you.
*Please check if the keywords are checked in the Mesh list.
*I would like the first sentence to be more concise. Now, it is lengthy to explain the various characteristics of the difficult work environment.
*Is the "Survey on the Status of Nursing at 31 Hospitals" a study of the country to which the author belongs? Or is it a worldwide survey? If it is a particular country, it is necessary to mention the name of the country. This should also be added to the title.
*Line 140~142 : Please check if the sentence is grammatically correct.
*Can't the two decimal places in Table 1 be written on the same line?
*Line 295: Why do you mention the improvement of 'perception' of the work environment? Does your tool include perception of your work environment?
*Line 306~326: It needs to be modified to increase the legibility of sentences and paragraphs.The same goes for line 327-352. It is difficult to understand clearly what the author wants to discuss.
Author Response

(The authors gave the same response as above.)
